# The accuracy of diagnostic indicators for coeliac disease: A systematic review and meta-analysis

Martha M. C. Elwenspoek[1,2]*, Joni Jackson[1,2], Rachel O'Donnell[2], Anthony Sinobas[3], Sarah Dawson[1,2], Hazel Everitt[4], Peter Gillett[5], Alastair D. Hay[2], Deborah L. Lane[6], Susan Mallett[7], Gerry Robins[8], Jessica C. Watson[2], Hayley E. Jones[2], Penny Whiting[2]

**1** The National Institute for Health Research Applied Research Collaboration West (NIHR ARC West), University Hospitals Bristol NHS Foundation Trust, Bristol, United Kingdom, **2** Population Health Sciences, Bristol Medical School, University of Bristol, Bristol, United Kingdom, **3** Bristol Medical School, University of Bristol, Bristol, United Kingdom, **4** Primary Care Research Centre, University of Southampton, Southampton, United Kingdom, **5** Paediatric Gastroenterology, Hepatology and Nutrition Department, Royal Hospital for Sick Children, Edinburgh, Scotland, United Kingdom, **6** Patient representative, Cambridge, United Kingdom, **7** Centre for Medical Imaging, University College London, London, United Kingdom, **8** Department of Gastroenterology, York Teaching Hospital NHS Foundation Trust, York, United Kingdom

* Martha.Elwenspoek@bristol.ac.uk

**Data Availability Statement:** All relevant data are within the manuscript and its Supporting Information files.

## Abstract

### Background

The prevalence of coeliac disease (CD) is around 1%, but diagnosis is challenged by varied presentation and non-specific symptoms and signs. This study aimed to identify diagnostic indicators that may help identify patients at a higher risk of CD in whom further testing is warranted.

### Methods

International guidance for systematic review methods were followed and the review was registered at PROSPERO (CRD42020170766). Six databases were searched until April 2021. Studies investigating diagnostic indicators, such as symptoms or risk conditions, in people with and without CD were eligible for inclusion. Risk of bias was assessed using the QUADAS-2 tool. Summary sensitivity, specificity, and positive predictive values were estimated for each diagnostic indicator by fitting bivariate random effects meta-analyses.

### Findings

191 studies reporting on 26 diagnostic indicators were included in the meta-analyses. We found large variation in diagnostic accuracy estimates between studies and most studies were at high risk of bias. We found strong evidence that people with dermatitis herpetiformis, migraine, family history of CD, HLA DQ2/8 risk genotype, anaemia, type 1 diabetes, osteoporosis, or chronic liver disease are more likely than the general population to have CD. Symptoms, psoriasis, epilepsy, inflammatory bowel disease, systemic lupus

**Funding:** The work is funded by a Health Technology Assessment Programme (NIHR129020). The researchers were hosted by the National Institute for Health Research (NIHR) Applied Research Collaboration West (NIHR ARC West). Our funder requires us to retain copyright of this manuscript because this work also needs to be written in a report which will be published by the funder (as NIHR HTA report). The funders had no role in study design, data collection and analysis, decision to publish, or preparation of the manuscript.

**Competing interests:** The authors have declared that no competing interests exist.

**Abbreviations:** CD, Coeliac disease; EMA, endomysial; HLA, human leukocyte antigen; IgA, immunoglobulin A; IgG, immunoglobulin G; tTG, tissue transglutaminase.

erythematosus, fractures, type 2 diabetes, and multiple sclerosis showed poor diagnostic ability. A sensitivity analysis revealed a 3-fold higher risk of CD in first-degree relatives of CD patients.

## Conclusions

Targeted testing of individuals with dermatitis herpetiformis, migraine, family history of CD, HLA DQ2/8 risk genotype, anaemia, type 1 diabetes, osteoporosis, or chronic liver disease could improve case-finding for CD, therefore expediting appropriate treatment and reducing adverse consequences. Migraine and chronic liver disease are not yet included as a risk factor in all CD guidelines, but it may be appropriate for these to be added. Future research should establish the diagnostic value of combining indicators.

## Introduction

Coeliac disease (CD) is underdiagnosed: the prevalence is estimated to be as high as 1%, but only around one in four cases are diagnosed [1,2]. CD is a chronic immune-mediated enteropathy occurring in genetically predisposed individuals and precipitated by exposure to dietary gluten from wheat, rye and barley, causing a variable degree of intestinal damage. In most patients, this will reverse on a gluten-free diet. However, due to the varied presentation of non-specific clinical signs and symptoms, recognising CD is difficult. Many CD patients experience a delay in diagnosis, especially when having non-specific symptoms, which can take several years [3]. If unrecognised and untreated, the accumulating damage in the small intestines impairs nutrient absorption which can lead to osteoporosis and anaemia, and increases the risk of developing pregnancy-related complications and certain types of cancer [4,5].

The first step in the diagnostic pathway is a serological test that measures immunoglobulin A (IgA) against tissue transglutaminase (tTG), endomysial (EMA), or deaminated gliadin peptide. In IgA deficient patients, IgG based tests such as tTG-IgG or EMA-IgA should be measured instead. Patients who are seropositive are usually required to have a confirmation biopsy, in which the histopathology of duodenal tissues is investigated for villous atrophy [6]. The European Society Paediatric Gastroenterology, Hepatology and Nutrition (ESPGHAN) guidelines [7] and European Society for the Study of Coeliac Disease (ESsCD) guidelines [8] suggest that biopsies can be avoided in children who have high tTG-IgA levels and a confirmatory EMA-IgA test with or without human leukocyte antigen (HLA) genotyping. Evidence supporting a biopsy avoidance strategy in adults is also accumulating.

CD can be treated effectively by lifelong elimination of gluten from the diet, which can reverse intestinal damage and prevent long-term consequences [9]. Because an effective treatment is available, clinical detection is difficult, and CD has a relatively high prevalence, it fulfils several WHO criteria for population screening [10,11]. However, mass testing may have associated harms such as medicalisation, patient anxiety, and unnecessary invasive biopsy. Screening "at risk" groups, on the other hand, appears to be a promising active case finding strategy to tackle underdiagnosis of CD [12] and is recommended by current guidelines [6–8]. Improved case finding will enable patients to start the diet as early as possible. However, the list of symptoms and risk conditions that should prompt serological testing varies between guidelines. In this systematic review we assess the relevance of various symptoms and risk

factors in "diagnosing" CD, considering the potential of these as initial screening tools prior to serological testing. We will refer to these as "diagnostic indicators".

## Methods

### Protocol and registration

The review was registered with at PROSPERO (CRD42020170766) and a protocol has been published [13]. We followed recommendations from the Centre for Reviews and Dissemination [14], the Cochrane Handbook for Systematic Reviews of Diagnostic Test Accuracy [15], and reported according to the PRISMA DTA statement [16].

### Eligibility criteria

Studies including adults and/or children with or without a potential diagnostic indicator who were all tested for CD with serological tests (tTG, EMA, or deaminated gliadin peptide IgA/IgG) and/or duodenal biopsy using a "single-gate" [17] (such as cross-sectional or cohort) or "multi-gate" (such as case control) design were eligible for inclusion. Studies were treated as diagnostic test accuracy studies, where the diagnostic indicator was treated as the index test and CD serological tests and/or biopsy as the reference standard. Studies were included if all participants were tested for CD, the control group was representative of the general population, and sufficient data could be extracted to construct cross-tabulations of the number of people with and without the diagnostic indicator against the number of people with and without CD (2x2 data). We excluded studies published before 1997 (the year in which tTG was developed), to reduce the variation in CD diagnostic tests. Prediction modelling studies were also eligible for inclusion. We did not apply restrictions on age or publication language.

We defined diagnostic indicators as signs, symptoms, or risk factors that may help clinicians identify patients in whom further testing for CD is warranted. We did not consider factors that are difficult to determine at an initial consultation, such as perinatal risk factors, age at gluten introduction, or experimental factors that are not measured in clinical practice (i.e. tests for susceptibility genes other than HLA-DQ status, which are currently not widely available to clinicians and therefore not (yet) useful in aiding diagnosis).

### Information sources

MEDLINE, Embase, Cochrane Library, and Web of Science were searched from 1997 until April 2021. Ongoing and completed studies were identified using the WHO International Clinical Trials Registry and the NIH Clinical Trials database.

### Search strategy

The search strategy incorporated three main elements: (1) conditions (CD) + prognostic/predictive research filter [18,19], (2) conditions (CD) + all physical diseases/signs/symptoms (based on MeSH, EMTREE) + 'CD' diagnosis, (3) terms for high risk populations (see Supplementary methods) [13]. Animal studies, case reports, letters, editorials, and coeliac artery/trunk research were filtered out and a sensitive study design filter was applied. We also screened reference lists of the latest guidelines on CD and recent systematic reviews.

### Study selection

We followed a two-staged study selection: (1) abstract screening stage, in which clearly irrelevant papers were excluded, (2) full text assessment, in which possibly relevant records identified in the initial screening were assessed in detail and reasons for exclusion were documented.

Both stages were performed independently by two reviewers with disagreements resolved through discussion or referral to a third reviewer.

## Data collection process

Data extraction was performed using standardized forms by one reviewer and checked by a second with disagreements resolved through discussion or referral to a third reviewer. We extracted the following data where reported: study and participant characteristics, details on the diagnostic indicator and CD diagnosis, and 2x2 data. Study populations were categorised as "children" if the majority were children and none of the participants were older than 21; and as "adults" if the majority were adults with no participant younger than 15. All other populations were categorised as a mixed age group. Diagnostic indicators were grouped based on discussion with clinical team members; for example, acid reflux symptoms included heartburn, dyspepsia, and gastroesophageal reflux symptoms. If more than one outcome was reported in one study, e.g. heartburn and dyspepsia, only one was included in the meta-analysis to avoid including the same individuals twice. In those cases, the broader term (e.g. dyspepsia over heartburn) or more prevalent diagnostic indicator (e.g. HLA-DQ2 over HLA-DQ8) was selected.

## Risk of bias

Risk of bias was assessed separately for each diagnostic indicator reported in a study, using the QUADAS-2 tool [20], which includes domains covering participants, index test, reference standard and flow and timing. If at least one of the domains was rated as "high risk" the study results were considered at high risk of bias; if all domains were judged as "low risk" the study was considered at low risk of bias, otherwise the study was considered at "unclear" risk of bias. The content of the tool was tailored to the review by making the following modifications to the QUADAS-2 risk of bias signalling questions: Due to the broad research question and the expected heterogeneity between included studies, the signalling questions about concerns regarding applicability were left out. We also took out two signalling questions for the index test and one for the reference standard ("Were the index test results interpreted without knowledge of the results of the reference standard?", "If a threshold was used, was it pre-specified?", "Were the reference standard results interpreted without knowledge of the results of the index test?"), which were considered not relevant because in this review the index test is not a test but diagnostic indicator and the reference standard is a diagnosis of CD. These index test questions were replaced by "Was the aim of the study to investigate this diagnostic indicator?". Risk of bias was assessed by one reviewer and checked by a second.

## Synthesis of results

For each diagnostic indicator, we fitted a bivariate random effects meta-analysis, assuming binomial likelihoods for numbers of true positives and true negatives in each study [21,22]. We reported summary estimates of sensitivity and specificity and their estimates of the between-study standard deviation on the logit scale ("tau"). Study-specific and summary estimates of sensitivity and specificity were presented per diagnostic indicator in coupled forest plots and summary receiver operating characteristic (ROC) plots with 95% confidence ellipses and summary ROC curves.

Summary results from each meta-analysis were also used to estimate positive predictive values (PPVs), i.e. the probability of CD given that an individual has each diagnostic indicator. To calculate these values, we assumed a prevalence of 1% of CD in the general population [23,24]. 95% confidence intervals (CIs) around PPVs were computed using Monte Carlo

simulation, simulating from a bivariate normal distribution for summary sensitivity and specificity on the logit scale. Negative predictive values are not informative in this context, because a sign, symptom, or risk condition cannot be used in clinical practice to exclude CD; these are therefore not reported.

### Sensitivity analyses and subgroup analyses

Because we expected heterogeneity across studies in sensitivity and specificity due to variability in age groups (children vs adults), method of CD diagnosis (biopsy and/or serology versus serology only), and study design (single-gate versus multi-gate), we performed subgroup and sensitivity analyses on these study characteristics if subgroups contained at least 5 studies.

All statistical analyses were performed in R version 4.0.2 [25].

### Deviations from the protocol

Due to the size of the review and time constraints, it was decided to not extract data on additional diagnostic indicators which were reported by fewer than 5 studies. We provide full references for all studies reporting on indicators for which we did not extract data. A post-hoc sensitivity analysis was performed on the diagnostic indicator 'family history of CD'.

### Patient and public involvement

The study was designed with two patient co-applicants who are 'experts by experience' being affected day to day by CD. As co-applicants for the project they contributed to provide input during the project proposal stage, attending project meetings to provide context from a patient viewpoint, and providing feedback on research materials to ensure relevance to patient interests.

## Results

The literature searches and reference lists of 22 systematic reviews and four recent guidelines on CD [7–9,26,27] identified 12,027 records after deduplication. We selected 709 records for full text assessment. 241 studies fulfilled the inclusion criteria containing 387 reports of 91 distinct diagnostic indicators. S1 Table provides a list of diagnostic indicators and references for which we did not extract data (due to fewer than five studies reporting on the indicator). In total, 191 studies reporting on 26 distinct indicators were included in our meta-analyses (Fig 1).

### Study characteristics

The included diagnostic indicators consisted of 7 symptoms, 17 risk conditions, and 2 genetic predispositions (see Table 1 for summary study characteristics; S2 Table for study-level details). Among symptoms, abdominal pain (n = 12) and diarrhoea (n = 12) were reported on by the highest number of studies; among risk conditions, most reported on were type 1 diabetes (n = 31) and thyroid disease (n = 23). Studies investigating symptoms associated with CD predominantly used a cohort or cross-sectional design, using a serological test to detect CD. Studies looking at risk conditions mainly used case-control designs, where people with CD were compared to a healthy control group. Most studies included adult participants, although many diagnostic indicators were also studied in a population of children or a mixed population. Although sample sizes for each meta-analysis ranged between 1,004 and 55,500

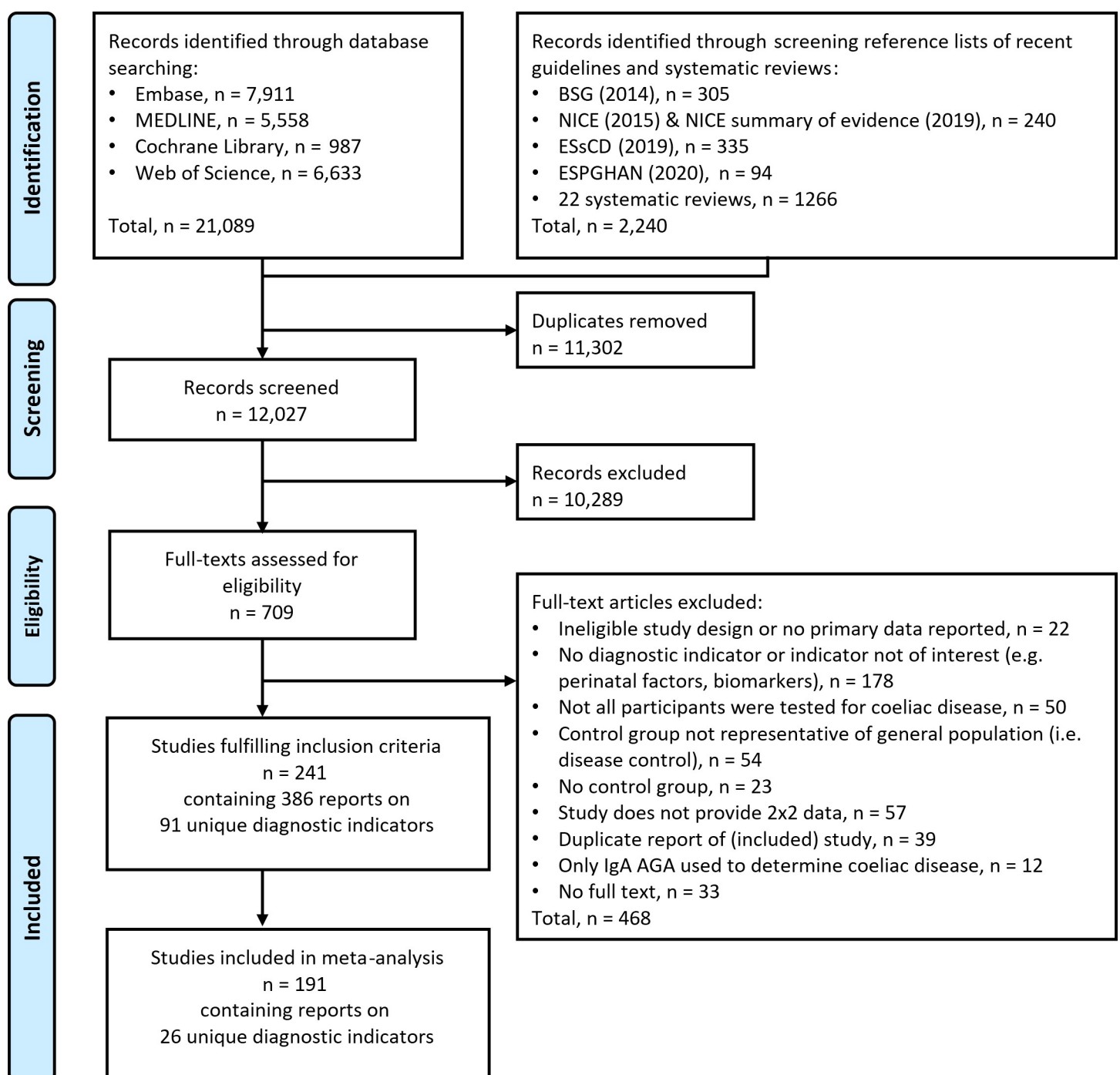

**Fig 1. PRISMA flow diagram.** Abbreviations: BSG: Guidelines from the British Society of Gastroenterology [9]; NICE: National Institute for Health and Care Excellence; ESsCD: European Society for the Study of Coeliac Disease guideline [8]; ESPGHAN: European Society Paediatric Gastroenterology, Hepatology and Nutrition Guidelines [7].

participants, some meta-analyses were based on a small number of CD patients, as prevalence was often low. For instance, for multiple sclerosis and systemic lupus erythematosus estimates of sensitivity are based on only 12 and 9 people with CD, respectively.

**Table 1. Summary table of study characteristics.**

| Diagnostic indicator | Diagnostic indicator details | Studies | Total sample | CD patients | Age groups | Study Designs | Control groups | Reference standards (CD diagnosis strategy) |
|---|---|---|---|---|---|---|---|---|
| **Symptoms** | | | | | | | | |
| Abdominal pain | (Recurrent or acute) abdominal or stomach pain | 12 | 48,451 | 1,014 | Adults, n = 6 Children, n = 6 | Case-control (DI),[1] n = 3 Nested case-control (CD), n = 2 Cohort/cross-sectional, n = 7 | Healthy controls, n = 3 Population sample without diagnostic indicator, n = 9 | Serology only, n = 8 Biopsy +/- serology, n = 4 |
| Acid reflux symptoms | Dyspepsia, functional dyspepsia, GERS, heartburn | 10 | 12,192 | 534 | Adults, n = 9 Mixed, n = 1 | Case-control (DI),[1] n = 3 Nested case-control (DI),[1] n = 2 Nested case-control (CD), n = 1 Cohort/cross-sectional, n = 4 | Healthy controls, n = 3 Population sample without diagnostic indicator, n = 7 | Serology only, n = 6 Biopsy +/- serology, n = 4 |
| Bloating or abdominal distension | Bloating, abdominal distension | 6 | 32,694 | 624 | Adults, n = 4 Children, n = 2 | Nested case-control (CD), n = 1 Cohort/cross-sectional, n = 5 | Population sample without diagnostic indicator, n = 6 | Serology only, n = 4 Biopsy +/- serology, n = 2 |
| Constipation | (Chronic) constipation | 12 | 54,286 | 943 | Adults, n = 5 Children, n = 7 | Case-control (DI),[1] n = 1 Nested case-control (DI),[1] n = 1 Nested case-control (CD),[2] n = 1 Cohort/cross-sectional, n = 9 | Healthy controls, n = 1 Population sample without diagnostic indicator, n = 11 | Serology only, n = 8 Biopsy +/- serology, n = 4 |
| Diarrhoea | Diarrhoea | 13 | 55,500 | 1126 | Adults, n = 7 Children, n = 6 | Case-control (DI),[1] n = 1 Nested case-control (CD),[2] n = 2 Cohort/cross-sectional, n = 10 | Healthy controls, n = 1 Population sample without diagnostic indicator, n = 12 | Serology only, n = 10 Biopsy +/- serology, n = 3 |
| Vomiting and nausea | Vomiting, nausea, nausea after eating | 7 | 44,937 | 435 | Adults, n = 3 Children, n = 4 | Cohort/cross-sectional, n = 7 | Population sample without diagnostic indicator, n = 7 | Serology only, n = 6 Biopsy +/- serology, n = 1 |
| Weight loss | Weight loss | 5 | 31,739 | | Adults, n = 3 Children, n = 2 | Nested case-control (CD), n = 2 Cohort/cross-sectional, n = 3 | Population sample without diagnostic indicator, n = 5 | Serology only, n = 4 Biopsy +/- serology, n = 1 |
| **Risk conditions** | | | | | | | | |
| Anaemia | IDA, low haemoglobin levels, pernicious anaemia, of obscure origin or unspecified | 17 | 13,477 | 715 | Adults, n = 13 Children, n = 4 | Case-control (DI),[1] n = 9 Nested case-control (CD),[2] n = 2 Cohort/cross-sectional, n = 6 | Healthy controls, n = 8 Population sample without diagnostic indicator, n = 9 | Serology only, n = 9 Biopsy +/- serology, n = 8 |
| Arthritis | RA, AS, juvenile idiopathic arthritis, PsA, juvenile rheumatic diseases | 15 | 10,745 | 542 | Adults, n = 8 Children, n = 5 Mixed, n = 2 | Case-control (DI),[1] n = 11 Nested case-control (CD)[2], n = 1 Cohort/cross-sectional, n = 3 | Healthy controls, n = 13 Population sample without diagnostic indicator, n = 2 | Serology only, n = 7 Biopsy +/- serology, n = 8 |
| Chronic liver disease | Hepatic disease, hepatitis, PBC, (unexplained) abnormal liver enzymes, ALD, chronic hepatitis C | 15 | 8,682 | 448 | Adults, n = 9 Children, n = 2 Mixed, n = 4 | Case-control (DI),[1] n = 12 Nested case-control (CD)[2], n = 1 Cohort/cross-sectional, n = 2 | Healthy controls, n = 12 Population sample without diagnostic indicator, n = 3 | Serology only, n = 7 Biopsy +/- serology, n = 8 |
| Dermatitis herpetiformis | Dermatitis herpetiformis | 5 | 1,429 | 579 | Adults, n = 4 Mixed, n = 1 | Case-control (DI),[1] n = 3 Nested case-control (CD)[2], n = 2 | Healthy controls, n = 3 Population sample without diagnostic indicator, n = 2 | Serology only, n = 4 Biopsy +/- serology, n = 1 |

(*Continued*)

**Table 1.** (Continued)

| Diagnostic indicator | Diagnostic indicator details | Studies | Total sample | CD patients | Age groups | Study Designs | Control groups | Reference standards (CD diagnosis strategy) |
|---|---|---|---|---|---|---|---|---|
| Epilepsy | Epilepsy, ataxia | 12 | 10,717 | 505 | Adults, n = 2 Children, n = 9 Mixed, n = 1 | Case-control (DI)[1], n = 11 Nested case-control (CD)[2], n = 1 | Healthy controls, n = 11 Population sample without diagnostic indicator, n = 1 | Serology only, n = 5 Biopsy +/- serology, n = 7 |
| Fracture | Vertebra fracture, wrist fracture, fractures (unspecified) | 8 | 24741 | 549 | Adults, n = 8 | Case-control (DI)[1], n = 3 Nested case-control (CD)[2], n = 1 Cohort/cross-sectional, n = 4 | Healthy controls, n = 3 Population sample without diagnostic indicator, n = 5 | Serology only, n = 7 Biopsy +/- serology, n = 1 |
| Inflammatory bowel disease | Ulcerative colitis, Crohn's disease | 6 | 2,886 | 32 | Adults, n = 4 Children, n = 1 Mixed, n = 1 | Case-control (DI)[1], n = 6 | Healthy controls, n = 6 | Serology only, n = 3 Biopsy +/- serology, n = 3 |
| Irritable bowel syndrome | Irritable bowel syndrome, functional gastrointestinal disorder | 18 | 18,446 | 842 | Adults, n = 17 Children, n = 1 | Case-control (DI)[1], n = 12 Nested case-control (DI)[1], n = 1 Nested case-control (CD)[2], n = 2 Cohort/cross-sectional, n = 3 | Healthy controls, n = 12 Population sample without diagnostic indicator, n = 6 | Serology only, n = 11 Biopsy +/- serology, n = 7 |
| Migraine | Migraine | 5 | 2,478 | 42 | Adults, n = 1 Children, n = 4 | Case-control (DI)[1], n = 5 | Healthy controls, n = 5 | Serology only, n = 2 Biopsy +/- serology, n = 3 |
| Multiple sclerosis | Multiple sclerosis | 5 | 1,086 | 12 | Adults, n = 4 Mixed, n = 1 | Case-control (DI)[1], n = 5 | Healthy controls, n = 5 | Serology only, n = 4 Biopsy +/- serology, n = 1 |
| Osteoporosis | Osteoporosis | 9 | 20,218 | 962 | Adults, n = 8 Mixed, n = 1 | Case-control (DI)[1], n = 4 Case-control (CD)[2], n = 1 Nested case-control (CD)[2], n = 2 Cohort/cross-sectional, n = 2 | Healthy controls, n = 4 Population sample without diagnostic indicator, n = 5 | Serology only, n = 6 Biopsy +/- serology, n = 3 |
| Psoriasis | Psoriasis | 6 | 1,127 | 44 | Adults, n = 3 Mixed, n = 3 | Case-control (DI)[1], n = 6 | Healthy controls, n = 5 Population sample without diagnostic indicator, n = 1 | Serology only, n = 4 Biopsy +/- serology, n = 2 |
| Subfertility or recurrent pregnancy loss | Idiopathic or immunologic infertility; previous or recurrent miscarriages, or implantation failure | 16 | 12,690 | 808 | Adults, n = 16 | Case-control (DI)[1], n = 12 Nested case-control (DI)[1], n = 1 Nested case-control (CD)[2], n = 2 Cohort/cross-sectional, n = 1 | Healthy controls, n = 13 Population sample without diagnostic indicator, n = 4 | Serology only, n = 12 Biopsy +/- serology, n = 4 |
| Systemic lupus erythematosus | Systemic lupus erythematosus | 6 | 1,004 | 9 | Adults, n = 5 Children, n = 1 | Case-control (DI)[1], n = 5 Cohort/cross-sectional, n = 1 | Healthy controls, n = 6 | Serology only, n = 2 Biopsy +/- serology, n = 4 |
| Thyroid disease | Autoimmune thyroid disease, Graves' disease, Hashimoto's thyroiditis, | 23 | 2,7031 | 1083 | Adults, n = 16 Children, n = 5 Mixed, n = 2 | Case-control (DI)[1], n = 15 Nested case-control (DI)[1], n = 2 Nested case-control (CD)[2], n = 2 Cohort/cross-sectional, n = 4 | Healthy controls, n = 15 Population sample without diagnostic indicator, n = 8 | Serology only, n = 13 Biopsy +/- serology, n = 10 |

(*Continued*)

**Table 1.** (Continued)

| Diagnostic indicator | Diagnostic indicator details | Studies | Total sample | CD patients | Age groups | Study Designs | Control groups | Reference standards (CD diagnosis strategy) |
|---|---|---|---|---|---|---|---|---|
| Type 1 Diabetes | Type 1 Diabetes | 31 | 26,635 | 1349 | Adults, n = 11 Children, n = 12 Mixed, n = 8 | Case-control (DI)[1], n = 28 Nested case-control (CD)[2], n = 1 Cohort/cross-sectional, n = 2 | Healthy controls, n = 27 Population sample without diagnostic indicator, n = 4 | Serology only, n = 17 Biopsy +/- serology, n = 14 |
| Type 2 Diabetes | Type 2 Diabetes | 6 | 8,199 | 110 | Adults, n = 4 Mixed, n = 2 | Case-control (DI)[1], n = 5 Cohort/cross-sectional, n = 1 | Healthy controls, n = 5 Population sample without diagnostic indicator, n = 1 | Serology only, n = 5 Biopsy +/- serology, n = 1 |
| **Genetic predisposition** | | | | | | | | |
| Family history of CD | Relatives with CD (first- or second-degree or unspecified) | 13 | 31,827 | 672 | Adults, n = 5 Children, n = 4 Mixed, n = 4 | Case-control (DI)[1], n = 6 Case-control (CD)[2], n = 1 Nested case-control (DI)[1], n = 1 Cohort/cross-sectional, n = 5 | Healthy controls, n = 6 Population sample without diagnostic indicator, n = 7 | Serology only, n = 12 Biopsy +/- serology, n = 1 |
| HLA DQ2/DQ8 | HLA DQ2 and/or HLA DQ8 | 9 | 19,466 | 513 | Adults, n = 1 Children, n = 6 Mixed, n = 2 | Case-control (DI)[1], n = 1 Case-control (CD)[2], n = 2 Nested case-control (DI)[1], n = 1 Nested case-control (CD)[2], n = 1 Cohort/cross-sectional, n = 4 | Healthy controls, n = 3 Population sample without diagnostic indicator, n = 6 | Serology only, n = 5 Biopsy +/- serology, n = 4 |

Abbreviations: DI: Diagnostic indicator; CD: Coeliac disease; GERS: Gastroesophageal reflux symptoms; IDA: Iron deficiency anaemia; RA: Rheumatoid arthritis; AS: Ankylosing spondylitis; PsA: Psoriatic arthritis; PBC: Primary biliary cirrhosis; ALD: Alcoholic liver disease.

1. (Nested) case-control (DI): (nested) case-control studies where cases were recruited based on having the diagnostic indicator. 'Nested' case-control studies are nested within a cohort, where cases and controls are selected from the same cohort.

2. (Nested) case-control (CD): (nested) case-control studies where cases were recruited based on having coeliac disease.

## Risk of bias

Several studies reported more than one diagnostic indicator, resulting in 290 risk of bias judgements. Most studies had methodological issues, and none were judged at low overall risk of bias (Fig 2; S1 Fig for risk of bias judgments per diagnostic indicator). In total, only 15 study reports were judged at a low risk of bias regarding patient selection. The main source of potential bias in patient selection was the use of a case-control study design. The index test domain was judged at low risk of bias if it was the study's main aim to investigate the diagnostic indicator of interest, which was the case for most studies. In total, 172 study reports on diagnostic indicators were judged at high risk of bias for reference standard. This was mainly driven by studies using serology tests without a confirmation biopsy to determine CD, which are therefore at risk of misallocating participants as CD patients or healthy controls. Flow and timing was judged at high risk of bias in studies that did not use the same combination of diagnostic tests for CD in all patients (reference standard); for example, in studies where biopsy was only performed in patients who had a positive serology test result.

## Accuracy of diagnostic indicators to detect coeliac disease

We found large variation in sensitivity, specificity, and PPV estimates between studies for most diagnostic indicators (Figs 3, S2 and S3, and S3 Table). Estimates of sensitivity were

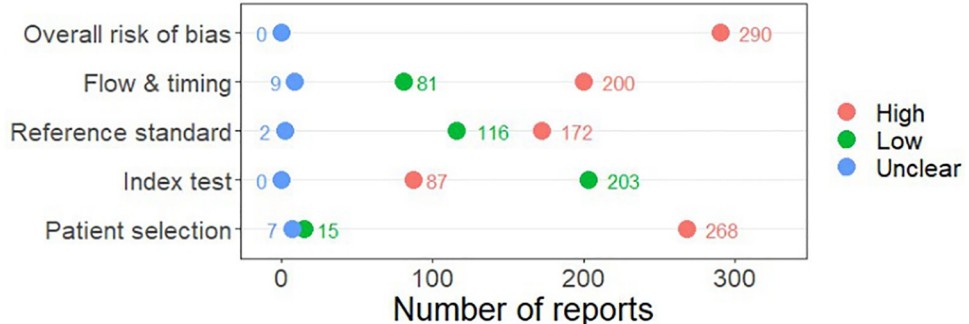

**Fig 2. Summary graph of risk of bias.**

particularly variable, often ranging from 0% to almost 100%, due to very small numbers of CD patients for some indicators.

The PPVs for the symptoms included in this review are similar to the baseline CD prevalence, suggesting that none of these symptoms provides additional diagnostic information (Fig 3, S3 Table). S3 Fig shows meta-analysis results in ROC space. A diagnostic indicator with a summary ROC curve closely following the diagonal line is no better at predicting CD than a coin toss, which is approximately the case for all symptoms.

Amongst risk conditions, dermatitis herpetiformis had the highest estimated sensitivity, specificity, and PPV (estimated PPV at 1% prevalence of CD = 29%, 95% CIs 3 to 72%). However, the uncertainty around these estimates was substantial, as is shown by the wide 95% CIs. We estimated PPVs above 2% for migraine, family history of CD, HLA DQ2/8, anaemia, type 1 diabetes, osteoporosis, and chronic liver disease. These estimates were relatively precise for HLA DQ2/8, anaemia, type 1 diabetes, osteoporosis, and chronic liver disease but there was considerable uncertainty for migraine and a family history of CD. People with thyroid disease, subfertility or recurrent pregnancy loss, or irritable bowel syndrome were 1.5–2 times more likely to have CD than the general population with 95% CIs lying entirely above the population prevalence of 1%. Although the estimated PPVs of psoriasis, epilepsy, inflammatory bowel disease, systematic lupus erythematosus, fracture, arthritis, and type 2 diabetes suggest an increased likelihood of CD in people with these conditions, there was considerable uncertainty in these estimates. The 95% CIs crossed or touched the line of population prevalence, indicating that the likelihood of CD may be similar to that in the general population. We found no evidence of an increased likelihood of CD in people with multiple sclerosis (Fig 3).

Similarly, arthritis, fracture, and type 2 diabetes appear to have no diagnostic ability when judging sensitivity and specificity in ROC space (S3 Fig). For multiple sclerosis, systemic lupus erythematosus, psoriasis, and inflammatory bowel disease there was not enough evidence to estimate a reliable summary ROC curve. For chronic liver diseases, epilepsy, migraine, irritable bowel syndrome, and dermatitis herpetiformis there was substantial uncertainty in summary estimates due to high variation between the study estimates. The summary ROC plots for type 1 diabetes, anaemia, subfertility or recurrent pregnancy loss, thyroid disease, and osteoporosis suggest a higher accuracy in predicting a CD diagnosis than a coin toss.

The HLA DQ2/8 risk genotype also had estimated PPVs above 2% (estimated PPV at 1% prevalence of CD = 2.6%, 95% CIs 2.2 to 3.1%). People with a family history of CD were not more likely to have CD than the general population and the summary ROC curve showed no diagnostic ability.

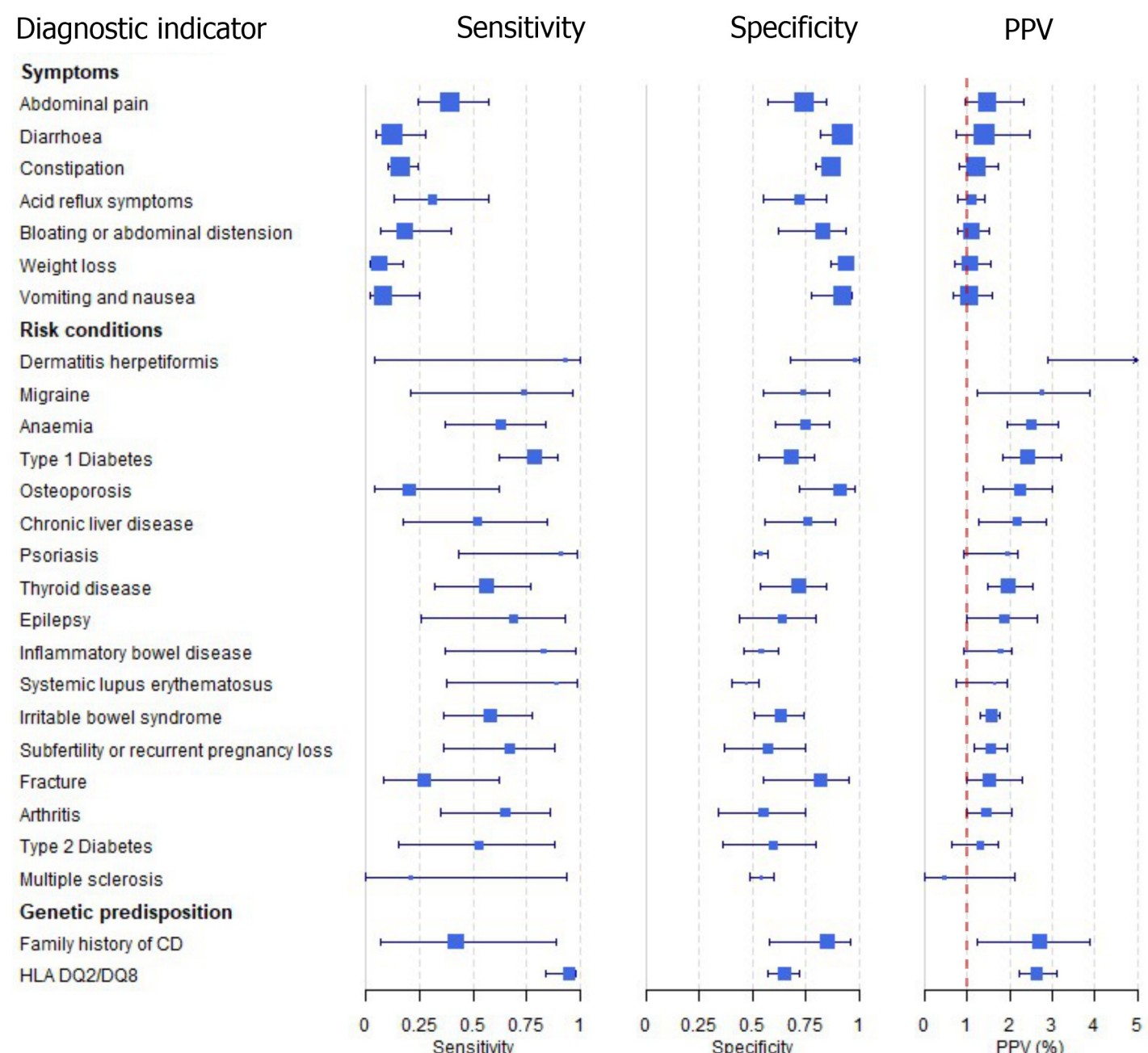

**Fig 3. Sensitivity, specificity, and positive predictive values.** Meta-analysis results are shown per diagnostic indicator. Positive predictive values (PPVs) were calculated for a population with a CD prevalence of 1% (red dotted line) using the estimated sensitivities and specificities from the meta-analyses. Diagnostic indicators are ordered from high to low PPV per diagnostic indicator group. The area of the box size is proportional to the total number of participants.

### Subgroup and sensitivity analyses

There were sufficient data on five diagnostic indicators to stratify the meta-analyses by age group (Fig 4, S4 Table). Estimated PPVs were similarly low and around 1% for abdominal pain, arthritis, constipation, and diarrhoea for adults and children. The results suggest that arthritis may be more predictive of CD in children than in adults, and abdominal pain and constipation may be more predictive for CD in adults that in children. The PPV of type 1

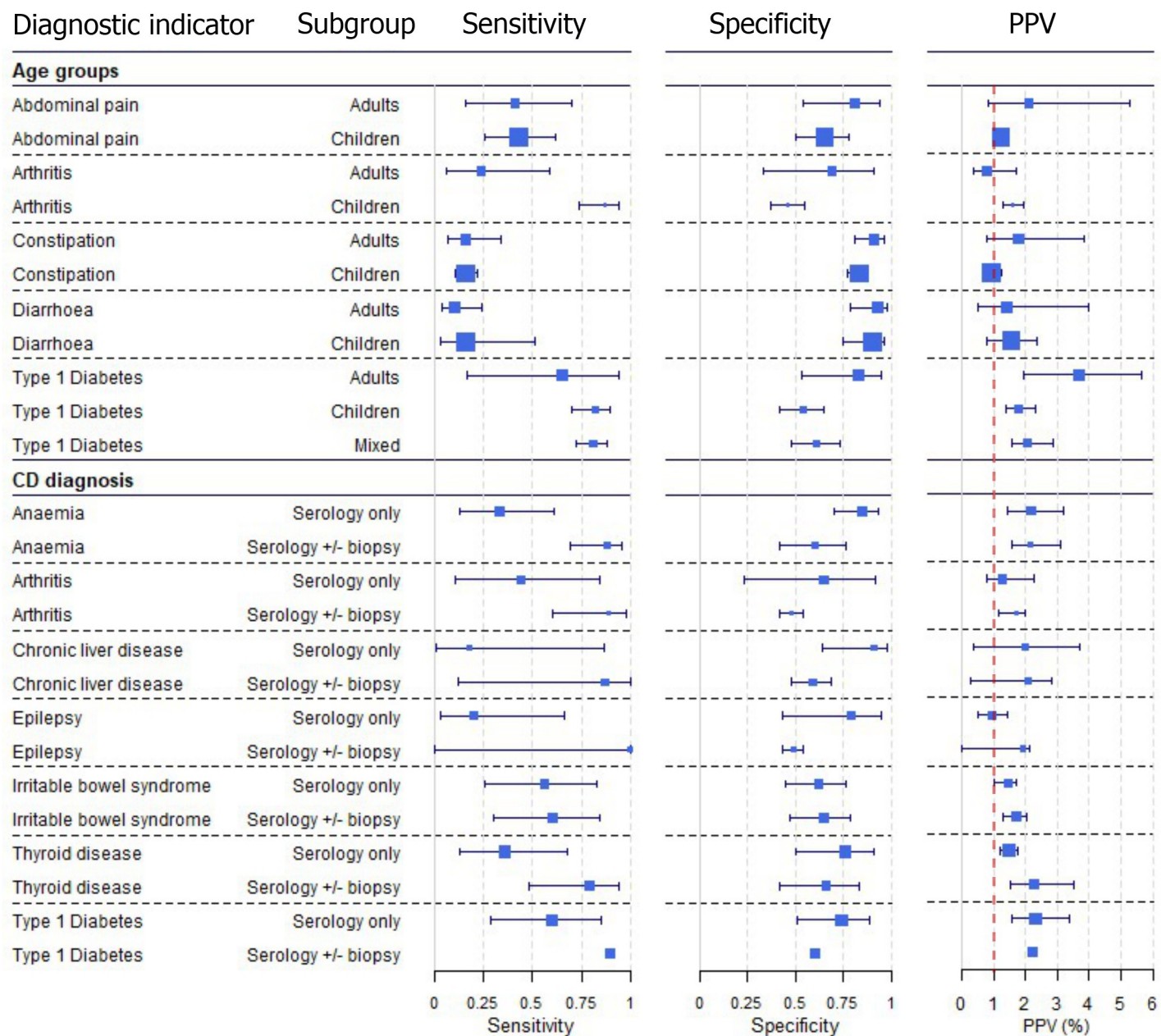

**Fig 4. Subgroup analysis stratified by age group and CD diagnosis.** Stratified meta-analysis results are shown per diagnostic indicator. Positive predictive values (PPVs) were calculated for a population with a CD prevalence of 1% (red dotted line) using the estimated sensitivities and specificities from the meta-analyses. The area of the box size is proportional to the total number of participants.

diabetes appeared higher for adults, estimated at 3.4% (95% CIs 1.9 to 5.6%), compared to children or mixed populations, at 1.8% (1.4 to 2.3%) and 2.1% (1.6 to 2.9%). However, each of these differences should be interpreted with caution since CIs overlap.

There were sufficient data on seven diagnostic indicators to stratify the analysis on CD diagnosis, comparing studies that used a serology-only approach and studies that included a conformation duodenal biopsy (Fig 4, S4 Table). Estimated PPVs were similar between the subgroups.

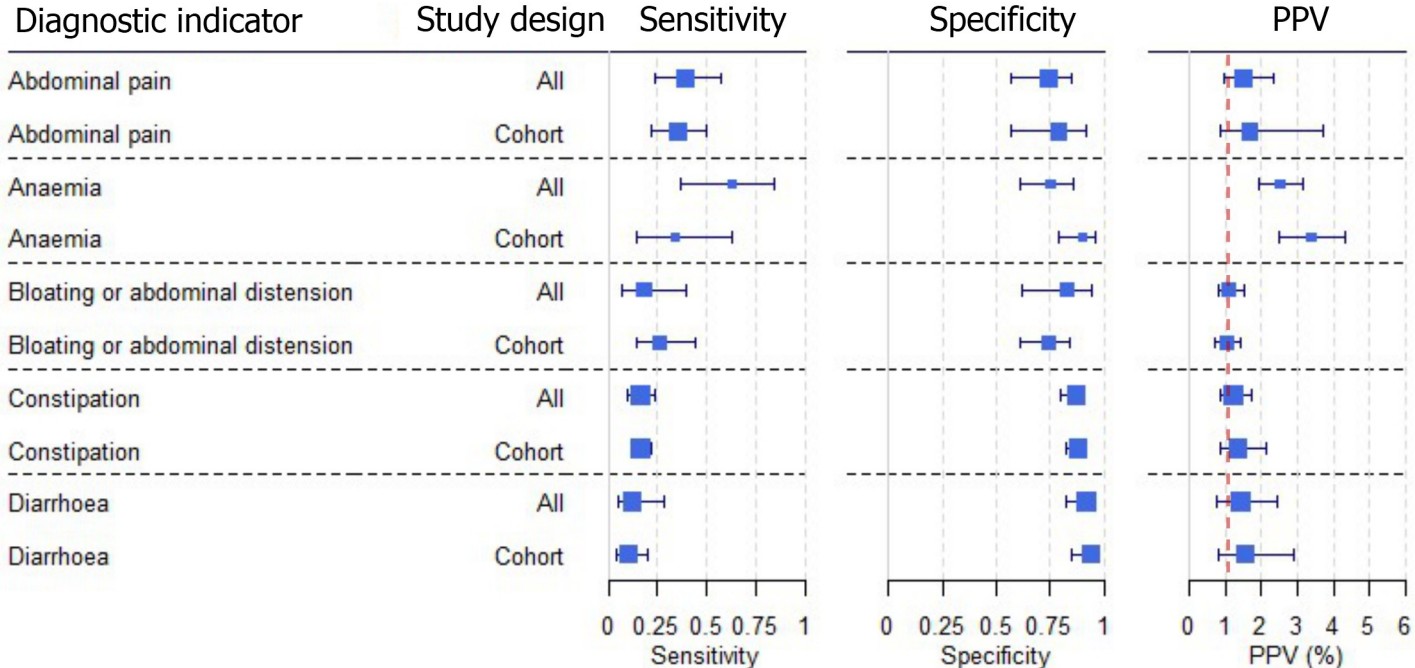

**Fig 5. Sensitivity analysis restricting to cohort studies.** Sensitivity meta-analysis on study design. Positive predictive values (PPVs) were calculated for a population with a CD prevalence of 1% (red dotted line) using the estimated sensitivities and specificities from the meta-analyses. The area of the box size is proportional to the total number of participants.

A sensitivity analysis was performed restricting to studies using a cohort or cross-section design for abdominal pain, anaemia, bloating or abdominal distension, constipation, and diarrhoea (Fig 5, S5 Table). Although case-control studies are more prone to bias than cohort studies, removing case control studies did not affect the sensitivity, specificity, or PPV estimates among these diagnostic indicators. It was not possible to perform a sensitivity analysis restricting to studies of low risk of bias, because all included studies were judged at overall high risk of bias.

Finally, a sensitivity analysis was performed on the diagnostic indicator 'family history of CD' restricting to studies that only included first-degree relatives. This increased the estimated PPV from 2.7% (95% CIs 1.2 to 3.9%) to 3.0% (1.6 to 3.7%), although note that CIs overlap.

## Discussion

### Principal findings

This systematic review summarises the accuracy of diagnostic indicators, such as symptoms and risk factors, to detect CD. Although none of the diagnostic indicators are accurate in "diagnosing" CD, some show promise in helping to identify patients who should be offered further testing. The estimated PPVs for migraine, family history of CD, HLA DQ2/8, anaemia, type 1 diabetes, osteoporosis, and chronic liver disease were all above 2%, with 95% CIs lying entirely above the population prevalence of 1%. In other words, people with these conditions are estimated to be more than twice as likely to have CD than the general population. However, for many indicators there was either little evidence or the studies were too heterogeneous to reliably estimate diagnostic accuracy. Dermatitis herpetiformis showed the highest diagnostic accuracy. However, because dermatitis herpetiformis is rare and treatment is a gluten-free

diet even in the absence of a CD diagnosis [28], it is unlikely to be helpful as a diagnostic indicator. Gastrointestinal symptoms showed poor diagnostic ability. The largest cohort studies included in our review found no significant differences in CD seroprevalence among individuals with and without specific gastrointestinal symptoms [29–31]. However, one cohort study found a higher prevalence of CD when multiple gastrointestinal symptoms were combined [30]. This suggests that, although any single symptom may not be useful in case finding it may be worth investigating the usefulness of combinations of symptoms.

All studies were judged at a high risk of bias, mainly due to use of serological tests to diagnose CD without a confirmatory biopsy, which may lead to an underestimation of the association between CD and a diagnostic indicator, or due to the use of a case control design, which may lead to an overestimation of the association. However, subgroup and sensitivity analyses showed no evidence of study design or method of CD diagnosis leading to an over- or underestimation of sensitivity, specificity, or PPV. A post hoc sensitivity analysis suggested that people with first-degree relatives with CD had a three times higher risk of CD than the general population.

## Strengths and limitations

We applied a robust methodological approach following internationally recognised systematic review guidance. We used a sensitive literature search strategy, and study selection was performed in duplicate. We applied stringent inclusion criteria to minimise bias. For instance, we only included studies where all participants had been tested for CD, which is important since CD is underdiagnosed.

The interpretation of the meta-analyses results is, nonetheless, limited by the substantial variability between studies. Although we investigated sources of variability by also performing stratified meta-analyses by age group, CD diagnosis, and study design, only a small minority of diagnostic indicators was reported by enough studies to perform these analyses. Another limitation is that our results, which are estimates of the accuracy of diagnostic indicators when used in isolation, cannot be used to estimate how predictive these indicators are in combination. Finally, we limited our review to diagnostic indicators that were reported by at least 5 studies; therefore, we may have missed other promising diagnostic indicators which are less often reported.

## Comparison with other studies

Our estimates of the probability of CD for people with certain risk conditions compared to the general population are in agreement with prevalence estimates of CD among individuals with those conditions. Meta-analyses estimated the prevalence of CD between 3–16% in people with type 1 diabetes [32], 1.6–3.8% (95% CIs) according to serology studies and 2.3–4.5% biopsy-proven CD in people with IBS [33], 2.6–3.9 among people with iron-deficiency anaemia [34], 1.6–2.6% in people with epilepsy [35], 1.3–1.9% in people with autoimmune thyroid disease [36], 1–7% in people with raised liver enzymes [37], and 1.1–2.0% in people with osteoporosis [38]. The odds of having CD was 1.7–2.7 higher in individuals with versus without psoriasis [39] and the risk of CD was 2.2–7.0 higher in patients with inflammatory bowel disease compared to controls [40]. The prevalence of CD in children with migraine-like headaches was estimated to be 1.5–3.7 times higher than in the general population, but no evidence was reported on adults populations [41]. Our analysis showed a similar increased risk of 2.8 fold, including one study with an adult population which showed similar results [42]. A population-based retrospective cohort study reported that among 160,000 patients headache-related visits,

including migraine, occurred 1.6 to 1.8 times more frequently in CD patients than in controls [43].

Recent data have suggested that infertility and CD are not associated [44,45], although most guidelines still name infertility as one of the risk factors for CD. Also, recent meta-analyses have shown conflicting results regarding the association between infertility and CD. Singh et al. (2016), Castaño et al. (2019), and Glimberg et al. (2021) reported a pooled prevalence of biopsy-proven CD in women with infertility between 1.4-3.5%, 0.6-2.8%, and 0.2-1.2%, respectively [46–48]. Glimberg et al. meta-analysed 11 studies and found a prevalence of CD among women with infertility similar to that of the general population [48]. Our results also suggest that women with subfertility have a 1.2 to 2-fold higher risk of CD. We used a broader definition of subfertility including recurrent pregnancy loss, whereas pregnancy loss was excluded by Glimberg et al., and we only included studies with control groups, which may explain the discrepancies in our and Glimberg's findings. Large scale prospective cohort studies, such as birth cohorts, are needed to address this question, where all participants are tested for CD and information on infertility is collected.

A meta-analysis of 6 studies investigated the accuracy of HLA-DQ2/DQ8 typing for the detection of CD and found a pooled sensitivity of 97–99% and specificity of 41–48% [49], compared to 86–99% and 56–71% in this review, respectively. Although almost 100% of individuals with CD are carriers of the HLA DQ2/DQ8 risk alleles, they only account for a small proportion of the heritability of CD [50]. We found a lower risk of CD in people with a family history of CD compared to other studies. A meta-analysis showed that the prevalence of CD is 6.3–8.8 times higher in first-degree relatives and 1.3–3.8 times higher in second degree relatives compared to the general population [51], whereas we found a risk of 1.3 times higher in people with a family history of CD. Only six of our included studies focussed specifically on first-degree relatives, whereas the other six included second-degree relatives or did not specify, which can partly explain our lower estimate. When restricting our analysis to first-degree relatives, we estimated the PPV at 1.3–7.2%. Finally, some studies included as few as 2 individuals with a family history of CD and 6 individuals with CD in their study population [52,53], which has likely attenuated the estimated association as well.

Small differences between our estimates of PPV and estimates from meta-analyses of prevalence may be explained by us only including studies that allowed estimation of both sensitivity and specificity (which requires some study participants to not have the diagnostic indicator).

## Implications for practice

Most of the promising indicators from our review, such as type 1 diabetes, thyroid disease, and osteoporosis, are recommended to prompt testing for CD by current guidelines. However, migraine, which had one of the higher estimated PPVs for CD, and chronic liver disease are not mentioned in most current guidelines. Future guidelines may want to recommend GPs to consider CD testing in patients with migraine or chronic liver disease.

There is a need for large cohort studies where all participants have received an accurate test for CD to reduce bias in estimates of the diagnostic ability of indicators such as symptoms or risk conditions. Accurate testing strategies that do not rely on invasive tests such as a duodenal biopsy would make this more feasible. Future research should investigate the accuracy of combinations of diagnostic indicators because single indicators with limited accuracy or low PPVs may still be valuable when used in combination. It is important that diagnostic prediction models use data in which all patients have been tested for CD to reduce bias as a result of underdiagnosis.

## Conclusion

Despite recent improvements in case finding, CD still represents a clinical iceberg with most cases yet to be detected. People with dermatitis herpetiformis, migraine, family history of CD, HLA DQ2/8 risk genotype, anaemia, type 1 diabetes, osteoporosis, or chronic liver disease have a higher risk of having CD than the general population and should be offered testing for CD. Migraine and chronic liver disease are not yet included as a CD risk factor in all testing guidelines, but it may be appropriate for this to be added. Symptoms showed limited predictive ability for CD. Future research should establish the diagnostic value of combining indicators. Combining multiple diagnostic indicators into prediction rules, especially if automated within electronic health records, may further improve case finding.

## Supporting information

**S1 PRISMA DTA checklist.**
(DOC)

**S1 Fig. Summary graph of risk of bias.**
(DOCX)

**S2 Fig. Forest plots of sensitivity and specificity.**
(DOCX)

**S3 Fig. Summary ROC plots per diagnostic indicators.**
(DOCX)

**S1 Table. List of diagnostic indicators not included in the meta-analysis.**
(DOCX)

**S2 Table. Study characteristics per indictor.**
(DOCX)

**S3 Table. Summary estimates of sensitivity, specificity, and prediction values.**
(DOCX)

**S4 Table. Summary estimates of sensitivity, specificity, and prediction values of subgroup analyses.**
(DOCX)

**S5 Table. Summary estimates of sensitivity, specificity, and prediction values of sensitivity analyses.**
(DOCX)

**S1 Search strategy.**
(DOCX)

## Acknowledgments

We would like to thank Jo Stubbs for her feedback from a patient perspective at the study design stage.

## Author Contributions

**Conceptualization:** Hazel Everitt, Peter Gillett, Alastair D. Hay, Deborah L. Lane, Susan Mallett, Gerry Robins, Jessica C. Watson, Hayley E. Jones, Penny Whiting.

**Data curation:** Martha M. C. Elwenspoek, Sarah Dawson.

**Formal analysis:** Martha M. C. Elwenspoek, Hayley E. Jones.

**Funding acquisition:** Martha M. C. Elwenspoek, Hazel Everitt, Peter Gillett, Alastair D. Hay, Deborah L. Lane, Susan Mallett, Gerry Robins, Jessica C. Watson, Hayley E. Jones, Penny Whiting.

**Investigation:** Martha M. C. Elwenspoek, Joni Jackson, Rachel O'Donnell, Anthony Sinobas, Hayley E. Jones, Penny Whiting.

**Methodology:** Sarah Dawson, Susan Mallett, Hayley E. Jones, Penny Whiting.

**Project administration:** Martha M. C. Elwenspoek, Joni Jackson.

**Resources:** Joni Jackson, Penny Whiting.

**Software:** Joni Jackson, Sarah Dawson.

**Supervision:** Martha M. C. Elwenspoek, Hayley E. Jones, Penny Whiting.

**Validation:** Martha M. C. Elwenspoek, Joni Jackson, Rachel O'Donnell, Hazel Everitt, Peter Gillett, Alastair D. Hay, Deborah L. Lane, Gerry Robins, Jessica C. Watson, Hayley E. Jones, Penny Whiting.

**Visualization:** Martha M. C. Elwenspoek, Hayley E. Jones.

**Writing – original draft:** Martha M. C. Elwenspoek.

**Writing – review & editing:** Joni Jackson, Rachel O'Donnell, Anthony Sinobas, Sarah Dawson, Hazel Everitt, Peter Gillett, Alastair D. Hay, Deborah L. Lane, Susan Mallett, Gerry Robins, Jessica C. Watson, Hayley E. Jones, Penny Whiting.

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
