## [Decision Letter · Decision Letter 0]

29 Sep 2021

The accuracy of diagnostic indicators for coeliac disease: a systematic review and meta-analysis

PONE-D-21-25925

Dear Dr. Elwenspoek,

We’re pleased to inform you that your manuscript has been judged scientifically suitable for publication and will be formally accepted for publication once it meets all outstanding technical requirements.

Kind regards,

Louise Emilsson

Academic Editor

PLOS ONE

Additional Editor Comments (optional):

Reviewers' comments:

Reviewer's Responses to Questions

**Comments to the Author**

1. Is the manuscript technically sound, and do the data support the conclusions?

Reviewer #1: Yes

Reviewer #2: Yes

2. Has the statistical analysis been performed appropriately and rigorously? 

Reviewer #1: Yes

Reviewer #2: Yes

3. Have the authors made all data underlying the findings in their manuscript fully available?

Reviewer #1: Yes

Reviewer #2: Yes

4. Is the manuscript presented in an intelligible fashion and written in standard English?

Reviewer #1: Yes

Reviewer #2: Yes

5. Review Comments to the Author

Reviewer #1: I was very happy with the responses of the authors. I am looking forward to seeing this paper online (and the health economic follow-up).

It is interesting that your update produced a significant association with liver disease. This coincides with the publication of another manuscript by Glimberg/Haggard et al, that found a 3.5% prevalence of celiac disease among individuals with autoimmune hepatitis, further supporting your findings.

https://pubmed.ncbi.nlm.nih.gov/34219350/

I also what to congratulation Penny Whiting for her earlier landmark paper on the QUADAS tool in BMC research methodology 18 years ago!

https://bmcmedresmethodol.biomedcentral.com/articles/10.1186/1471-2288-3-25

Reviewer #2: Although the term "diagnostic indicators" it is fully defined in the methods part, the use of the term already in the title is little bit confusing/ misleading as in the current "celiac disease language field" this usually refer to laboratory testing and diagnostic markers. No need the Title to be changed, but indeed, when reading the title I expected it will all be about biological and immunological indicators.

The paper is otherwise very interesting and well conducted; the results actually provide "scientific back-up" of an already assumed fact from clinical practice: the low predictive ability of the symptoms to detect CD.

6. PLOS authors have the option to publish the peer review history of their article (what does this mean?). If published, this will include your full peer review and any attached files.

Reviewer #1: No

Reviewer #2: No

---

## [Editor Report · Acceptance letter]

11 Oct 2021

PONE-D-21-25925 

The accuracy of diagnostic indicators for coeliac disease: a systematic review and meta-analysis 

Dear Dr. Elwenspoek:

I'm pleased to inform you that your manuscript has been deemed suitable for publication in PLOS ONE. Congratulations! Your manuscript is now with our production department. 

Kind regards, 

on behalf of

Dr. Louise Emilsson 

Academic Editor

PLOS ONE